# Urinary *Eubacterium* sp. *CAG:581* Promotes Non-Muscle Invasive Bladder Cancer (NMIBC) Development through the ECM1/MMP9 Pathway

**DOI:** 10.3390/cancers15030809

**Published:** 2023-01-28

**Authors:** Yuhang Zhang, Wenyu Wang, Hang Zhou, Yimin Cui

**Affiliations:** 1Institute of Clinical Pharmacology, Peking University First Hospital, Beijing 100034, China; 2Department of Pharmacy, Peking University First Hospital, Beijing 100034, China; 3Department of Gastroenterology, Beijing Friendship Hospital, Capital Medical University, Beijing 100054, China; 4Department of Urology Surgery, Beijing Luhe Hospital, Capital Medical University, Beijing 100054, China

**Keywords:** non-muscle invasive bladder cancer (NMIBC), *Eubacterium* sp. *CAG:581*, extracellular matrix protein 1 (ECM1), matrix metalloproteinase 9 (MMP9), noninvasive urinary diagnostics

## Abstract

**Simple Summary:**

The link between bladder cancer and urinary microbiota has been of high interest, but few studies have shed the light on the concrete urinary species as the potential clinical biomarkers. In this article, we reported the previously ignored urinary species *Eubacterium* sp. *CAG:581* in predicting the occurrence of non-muscle invasive bladder cancer (NMIBC). Excluding the favorable impact of age, sex, smoking, alcohol consumption and other potential contributing factors, we have found *Eubacterium* sp. *CAG:581* has contributed to the development of bladder cancer through activating the extracellular matrix protein 1-matrix metalloproteinase 9 pathway. In the cohort of 406 NMIBC patients and 398 healthy controls, we found that high *Eubacterium* sp. *CAG:581* will correctly distinguish an NMIBC patient from a healthy person by the chance of 79% within 3 years. Collectively, this study manifested that *Eubacterium* sp. *CAG:581* may serve as the promising noninvasive diagnostic biomarker for NMIBC.

**Abstract:**

Background: Increasing evidence points to the urinary microbiota as a possible key susceptibility factor for early-stage bladder cancer (BCa) progression. However, the interpretation of its underlying mechanism is often insufficient, given that various environmental conditions have affected the composition of urinary microbiota. Herein, we sought to rule out confounding factors and clarify how urinary *Eubacterium* sp. *CAG:581* promoted non-muscle invasive bladder cancer (NMIBC) development. Methods: Differentially abundant urinary microbiota of 51 NMIBC patients and 47 healthy controls (as Cohort 1) were first determined by metagenomics analysis. Then, we modeled the coculture of NMIBC organoids with candidate urinary *Eubacterium* sp. *CAG:581* in anaerobic conditions and explored differentially expressed genes of these NMIBC tissues by RNA-Seq. Furthermore, we dissected the mechanisms involved into *Eubacterium* sp. *CAG:581* by inducing extracellular matrix protein 1 (ECM1) and matrix metalloproteinase 9 (MMP9) upregulation. Finally, we used multivariate Cox modeling to investigate the clinical relevance of urinary *Eubacterium* sp. *CAG:581* 16S ribosomal RNA (16SrRNA) levels to the prognosis of 406 NMIBC patients (as Cohort 2). Results: *Eubacterium* sp. *CAG:581* infection accelerated the proliferation of NMIBC organoids (*p* < 0.01); ECM1 and MMP9 were the most upregulated genes induced by the increased colony forming units (CFU) gradient of *Eubacterium* sp. *CAG:581* infection via phosphorylating ERK1/2 in NMIBC organoids of Cohort 1. Excluding the favorable impact of potential contributing factors, the ROC curve of Cohort 2 manifested its 3-year AUC value as 0.79 and the cut-off point of *Eubacterium* sp. *CAG:581* 16SrRNA as 10.3 (delta CT value). Conclusion: Our evidence suggests that urinary *Eubacterium* sp. *CAG:581* promoted NMIBC progression through the ECM1/MMP9 pathway, which may serve as the promising noninvasive diagnostic biomarker for NMIBC.

## 1. Introduction

Bladder cancer (BCa) is the second-most common malignancy of the urinary tract, accounting for 4.6% of all new cancer diagnoses, with 400,000 new diagnoses and 160,000 deaths worldwide each year. Bladder cancer is age-associated, with individuals aged 75–84 years accounting for the largest percentage, at 30% of new cases per year [1]. BCa can be divided into muscle-infiltrating bladder cancer (MIBC) and non-muscle infiltrating bladder cancer (NMIBC) according to whether or not it invades into the muscular layer of the bladder wall [2]. NMIBC comprises approximately 75–80% of all newly diagnosed BCa [3], which is the malignant papillary tumor of the bladder confined to the mucosal layer (TIS/CIS accounting for 10%, Ta for 70%) and lamina propria (T1 for 20%) of the bladder, with no muscle infiltration [4]. After transurethral resection of the bladder tumor (TURBT), the 5-year risk of recurrence for high-risk NMIBC can be as high as 80% [5]. However, if NMIBC is screened at an early stage, patients with high-risk NMIBC have much lower rates of tumor recurrence after immediate postoperative perfusion, which can be delivered with simple surgery [6]; thus, improving the sensitivity and specificity of NMIBC screening is critical to the prognosis of NMIBC patients. Currently, the preferred method for screening patients with NMIBC is cystoscopy. Still, it is invasive and often screens to confirm for NMIBC only after the solid tumor has already been detected [7]. Exploring more convenient and sensitive early-stage screening modalities is essential for NMIBC treatment.

The development of NMIBC is a complex, multifactorial, and multistep pathological process in which both intrinsic genetic factors and extrinsic environmental factors have contributed to the process [8]. Regional, ethnic, and gender differences could affect the development of bladder cancer, with a high incidence at 50–70 years old. According to the National Central Cancer Registry of China (NCCRC), the ratio of male to female mortality is 2.97:1 [9,10]. Smoking is the most definite and major external risk factor for NMIBC. About 50% of NMIBC patients have a history of smoking, which increases the risk of bladder cancer by 2–5 times, depending on the intensity and duration of smoking [9,10]. Recent studies have shown the composition and content of the urinary flora can also impact the development and prognosis of NMIBC. Previous studies have indicated significant differences in urinary flora characteristics between BCa patients and healthy participants [11,12]. Based on their bioinformatics analysis, there may be a link between urinary flora and bladder cancer, and some different genera might be novel biomarkers for BCa [12]. However, these findings were premature, and the causal relationship between urinary flora and early-stage BCa is still not clear. It requires further validation through prospective cohort studies and further mechanistic explanation.

ECM1 is a glycoprotein, expressed in epithelial organs and secreted into the extracellular matrix, which can promote the proliferation of vascular endothelial cells and angiogenesis [13]. It has been found that the expression of ECM1 may be associated with BCa progression, which was confirmed by the potential anti-tumor effects observed in siRNA gene silencing experiments [13,14]. The overexpression of ECM1 has now been incorporated into scoring models to suggest poor clinical prognosis and metastatic potential in several types of cancer patients [15]. Matrix metalloproteinases (MMPs) are critical regulators of the tumor microenvironment. Their protein hydrolytic activity initiates the degradation of extracellular matrix components so that tumors are no longer restricted by the intact basement membrane, but are able to invade the surrounding tissues, ultimately leading to tumor malignancy [16]. MMPs have been shown to regulate not only the epithelial-mesenchymal transition through various non-catalytic structural domains, but also molecular signaling for cell growth, inflammation, and angiogenesis in a non-protein hydrolytic manner [17]. Previous studies have demonstrated that ECM1 inhibited MMP9 activity through high-affinity protein–protein interactions [17,18]. In certain cancer types, the significant increase in MMP9 transcription can be observed in parallel with the upregulation of ECM1, suggesting that the ECM1–MMP9 axis is of great significance for tumorigenesis and progression [17,19].

In this study, we have designed two clinical cohorts for prediction and validation, respectively. First, we compared the urinary microbiota of 51 NMIBC patients and 47 healthy controls, as Cohort 1, using metagenomic analysis. Then, we co-cultured NMIBC organoids with the candidate urinary bacterium *Eubacterium* sp. *CAG:581* and discovered by its underlying mechanisms that *Eubacterium* sp. *CAG:581* promoted the growth of NMIBC organoids by upregulating the expression of the ECM1–MMP9 axis, suggesting that *Eubacterium* sp. *CAG:581* may serve as the potential diagnostic predictor of NMIBC. To further verify whether *Eubacterium* sp. *CAG:581* has the similar predictive value for NMIBC in different large-scale populations, we designed Cohort 2 with 406 NMIBC patients and 398 healthy controls to determine the AUC and cut-off value of urinary *Eubacterium* sp. *CAG:581* 16SrRNA detection for potential clinical application.

## 2. Materials and Methods

### 2.1. Clinical Cohort Designs and Specimens

We studied two cohorts of NMIBC patients from Peking University First Hospital between 2019–2022. Patients were selected based on the availability of a pre-treatment urine sample and the presence of the well-modeling NMIBC organoids. On the one hand, urine samples and formalin-fixed paraffin-embedded tissues (FFPE) of 51 NMIBC patients and 47 healthy controls were collected as Cohort 1 (Detailed information was listed in Appendix A). We performed metagenome sequencing studies for the urine samples of Cohort 1 to define which bacterium was predominant in the urine of the NMIBC patients as compared to that of the healthy controls. Then, their FFPE were isolated with RNA for RNA-sequencing. On the other hand, we used the urine samples of 406 NMIBC patients and 398 healthy controls in Cohort 2 (Detailed information was listed in Appendix A) as a validation dataset to determine which levels of urinary *Eubacterium* sp. *CAG:581* generated from Cohort 1 could be applied and validated. Thus, Cohorts 1 and 2 serve as the models for our research discovery and validation, respectively.

All of the patient information is found in Appendix A. The Ethics Committees at Peking University First Hospital approved the study protocols (Ethical number: 2019–138). Written informed consent was obtained from all participants in this study.

### 2.2. Metagenome Sequencing

We collected urine samples primarily by collecting midstream clean capture urine (CC) and microbiologically cultured the urine using enhanced quantitative urine culture (EQUC) by inoculating 100 μL of urine onto 5% CO blood agar plates, mucin-nalidixic acid agar plates, and MacConkey agar plates for 248 h, followed by sequencing using 16S rRNA amplicons. All samples were then normalized to 0.2 ng/μL DNA material per library using a Quant-iT picogreen assay system (Life Technologies, Waltham, MA, USA, P11496) on an AF2200 plate reader (Eppendorf, Hamburg, Germany), then fragmented and tagged via tagmentation. Amplification was performed by Veriti 96-well PCR (Applied Biosystems 7500, Waltham, USA), followed by AMPure XP bead cleanup (Beckman Coulter, Brea, CA, USA). Fragment size was measured using Labchip GX Touch high-sensitivity. For cluster generation and next generation sequencing, samples were normalized to 1 nM, pooled, and diluted to 8 pM. The paired-end cluster kit V4 was used, and cluster generation was performed on an Illumina cBot, with pooled samples in all 8 lanes. Sequencing was performed on an Illumina HiSeq 2500 using SBS kit V4 chemistry. The median cluster densities were 908.5 for Nextera XT.

LEfSe scores measure the consistency of differences in relative abundance between taxa in the groups analyzed, with a higher core indicating higher consistency. We considered taxa with a linear discriminant analysis score >2 and *p* < 0.05 to be significant [20]. To identify species represented by the genera revealed by LEfSe, we first identified the OTUs associated with those genera, filtered out the low abundance OTUs (<50 copies), performed Kruskal–Wallis tests on each remaining OTU, and used BLAST (https://blast.ncbi.nlm.nih.gov) to align the sequences of these OTUs against the Greengenes database, retaining species with an identity match of >97% [21]. The BLAST algorithm was also used to query the predicted genes against the Kyoto Encyclopedia of Genes and Genomes (KEGG) database (http://www.genome.jp/kegg/, accessed on 15 June 2022), and the corresponding biological pathways were determined based on the obtained KEGG Orthology (KO) numbers. Gene Ontology (GO, http://www.geneontology.org/, accessed on 22 July 2022) annotations of the BLAST results were analyzed using blast2go [22]. The genomic circle map was constructed using Circos v0.64 (http://circos.ca/, accessed on 8 August 2022).

### 2.3. NMIBC Organoids Were Cocultured with Urinary Bacterium in the 2-Chamber Culture System

Surgically resected tissues were obtained from patients diagnosed with NMIBC in Cohort 1, which were collected endoscopically by cold cup biopsy. Then, all NMIBC tissue samples were placed on ice in the advanced Dulbecco’s Modified Eagle Medium/Nutrient Mixture (DMEM) F-12 medium and digested for 45 min at 37 °C. The remaining tissue aggregates were further digested in 4 mL of TrypLE Express recombinant enzyme (Invitrogen) for another 5 min at 37 °C. Subsequently, the suspension was filtered through a 70 μm nylon cell strainer and centrifuged for 5 min at 300× *g*. The tissue pellets were suspended in cold Matrigel, and the Matrigel cell suspension was seeded in prewarmed 6-well culture plates for passaging. Each organoid line was used from passages 5 to 12 (in most cases, at passage 8 or 9), so that lines displaying phenotypic instability had already completed their shift to a basal phenotype.

The plug of the coculture system fits tightly, physically blocking the influx of external oxygen, which allows for the maintenance of hypoxia in the basal chamber, while oxygen freely perfuses the apical chamber. NMIBC organoids were established on the array chip using 50% Matrigel. Each chip contained a reservoir layer on the top, a 3D implanting hole in the middle, and an anaerobic culture slide underneath. The nested design allowed for convenient medium exchange without disruption of the 3D organoids. After the coating process continued for 3 days, or when the layer was close to confluence, the medium in the basal chamber was equilibrated with anaerobic gas and subsequently sealed by inserting a plug made of butyl rubber (AsONE international, Santa Clara, CA, USA). The oxygen concentration of the apical chamber was measured with a fiberoptic oxygen meter (PreSens. Regensburg, Germany), which was set with 0 mg/L–0.2 mg/L oxygen. Then, we added urinary bacterium to the basal side in a medium suitable for exposure. For multiplicity of infection (MOI) calculations, a representative well can be harvested and cells counted, as described above.

### 2.4. RNA-seq and Data Processing

Total RNA was isolated from the FFPE of Cohort 1 using the RNeasy Micro Kit, then they were converted to cDNA with poly A primers using a TruSeq RNA Sample Preparation kit v2 (Illumina). FPKM (fragments per kilobase of transcript per million mapped fragments) was calculated as the gene expression level using Cufflinks version 2.2.1. High-throughput sequencing for mRNA-Seq was carried out using a Hiseq2500 (Illumina) system. For analysis and visualization of the data generated by Cufflinks, we used the DEseq R package to perform a differential analysis of gene expression, and the screening criteria were set as |log2(Fold change)| > 1 and *p*-value< 0.05.

### 2.5. Western Blotting

All NMIBC organoids were washed twice with cold PBS and harvested for Western blotting. Then, they were lysed in cold radioimmunoprecipitation assay (RIPA) buffer (50 mm Tris-HCl, pH 7.4, 150 mm NaCl, 1% Triton X-100, 1% sodium deoxycholate, 0.1% SDS) containing protease inhibitors for 30 min. The lysates were then centrifuged, and the supernatants were collected. Approximately 40 µg of total protein was denatured and separated by 10% SDS-PAGE, and then transferred to a polyvinylidene fluoride membrane. The membranes were blocked with 5% non-fat milk in Tris-buffered saline containing 0.1% Tween-20 (TBST) for 2 h at room temperature. The membranes were then incubated with primary antibodies overnight at 4 °C. The primary antibodies are listed in the Appendix A. β-actin was used as a loading control. The membranes were washed five times with TBST and then incubated with horseradish peroxidase-conjugated secondary antibodies for 1 h at room temperature. The signal was visualized by the LI-COR Odyssey^®^ Imaging System and assessed by Odyssey^®^ software (LI-COR Biosciences, Lincoln, NE, USA). All primary images (Appendix A) were cropped for presentation and quantified for statistical analysis (Appendix A).

### 2.6. Quantitative Real-Time PCR

Total RNA was isolated from the urinary microbiome with TRIzol (Invitrogen, Waltham, MA, USA). HiScript II qRT Super Mix (Vazyme, Nanjing, China) was used to synthesize the first strand of cDNA. Quantitative real-time PCR was performed using the Hieff qPCR SYBR Green PCR Master Mix (Yeasen, Shanghai, China) on the ABI 7300 TH Real-Time PCR System (Applied Biosystems). A total of 10 µL of dye, 1 µL of upstream and downstream primer, 1 µL of dNTP, 2 µL of Taq polymerase, 5 µL of cDNA of the sample to be tested, and 30 µL of ddH_2_O were mixed and centrifuged briefly at 6000 rpm for PCR amplification reaction. The reaction conditions were: 2 min pre-denaturation at 93 °C, then 40 cycles at 93 °C for 1 min, 55 °C for 1 min, 72 °C for 1 min, and finally, a 7 min extension at 72 °C. The primers of *Eubacterium* sp. *CAG:581* 16SrRNA, *ptpn6*, *ikzf3*, *ecm1*, *mmp9*, and *gapdh* are listed in the Appendix A. All target gene transcripts were normalized to *GAPDH*, and the relative fold change in the expression was calculated using the 2^−ΔΔCT^ method [23].

### 2.7. Adenoviral shRNA Infection of NMIBC Organoids

The shRNA adenoviruses Ad-GFP-U6-m-ECM1-shRNA (shECM1, CCTGATATTTCCTCGGGTCTT) were purchased from Vector Biolabs. A non-specific scrambled shRNA adenovirus Ad-GFP-U6-scrambled-shRNA (Vector Biolabs, 1122N, CCGGCAACAAGATGAAGAGCACCAACTCGAGTTGGTGCT), expressing green fluorescent protein (GFP) alone, was used as a control. For adenoviral infection, after centrifugation of dissociated NMIBC organoids in 3.5% BSA gradient, the resulting pellet was resuspended in serum-free DMEM medium containing Ad-GFP-U6-m-ECM1-shRNA or Ad-GFP-U6-scrambled-shRNA at a concentration of 2 × 10^8^ pfu/mL. The mixture was then plated on a normoxic basal chamber coated with poly-L-lysine/laminin and incubated at 37 °C in 5% CO_2_. After 2 h of incubation, the medium was replaced with fresh supplemented DMEM medium. A total of 3 days after infection, adenovirus-infected NMIBC organoids were used for protein extraction to determine the expression of ECM1. When passaging to the 4–8th generation, BCa organoid^shECM1^ and control BCa organoid^vector^ were exposed to 10 μM ravoxertinib (Rav) or ulixertinib (Uli) coculture for 24 h, to identify downstream pathway activation, and 21 days of 10^7^ cfu *Eubacterium* sp. *CAG:581* of proliferation analysis. SCH772984, ravoxertinib and LY3214996, ulixertinib were both purchased from Selleckchem.

### 2.8. Statistical Analysis

Statistical analyses were carried out using the program R (www.r-project.org, accessed on 13 May 2022). Data from at least three independent experiments performed in triplicate are presented as the means ± SD. Error bars in the scatterplots and the bar graphs represent SD. Data were examined to determine whether they were normally distributed with the one-sample Kolmogorov–Smirnov test. If the data were normally distributed, comparisons of measurement data between two groups were performed using the independent sample *t* test, and the comparisons among three or more groups were applied with the S-N-K (Student–Newman–Keuls) post hoc test in the analysis of ANOVA. When the data represented a skewed distribution, comparisons were performed by nonparametric testing. Measurement data between the two groups were compared using the nonparametric Mann–Whitney test, the Pearson correlation, and Chi-square tests of independence for the correlation analyses.

To generate the ROC curves, patients were classified as having a recurrence time either longer or shorter than the median recurrence free survival period, excluding patients who were alive for durations less than the median recurrence free survival time at the last follow-up. Single-sample gene set enrichment analysis (ssGSEA) was used to assess gene set activation scores in the gene expression profiling data. ssGSEA calculates a sample level gene set score by comparing the distribution of gene expression ranks inside and outside the gene set. The ssGSEA score was calculated by gene set variation analysis (GSVA) in R package. GraphPad Prism v.8.01 was also used for statistical analysis (GraphPad Software, La Jolla, CA, USA). Univariate and multivariate Cox regression models were used to evaluate the occurrence of NMIBC. The hazard ratio (HR) and 95 percent confidence interval (CI) were adopted to find key factors correlated with NMIBC occurrence.

## 3. Results

### 3.1. Eubacterium *sp.* CAG:581 Is Clinically Associated with NMIBC Occurrence

To examine the potential relationship between the urinary microbiota alteration and NMIBC development, we first compared the long-read metagenomics sequencing data of 51 NMIBC patients and 47 healthy controls. The LEfSe algorithm was used to define their potential differential bacterium patterns. We found that *Eubacterium* sp. *CAG:581*, *Bacteroides* sp. *4_3_47FAA*, and *Flavobacteriales* were enriched in NMIBC group as compared to the healthy control group (Figure 1A). *Flavobacteriales* belong to the taxonomy level of class, hence we further studied *Eubacterium* sp. *CAG:581* and *Bacteroides* sp. *4_3_47FAA* for quantitative validation. The gut microbiota diversity of the above two groups was analyzed by the Chao1 index and rank abundance curves for α diversity and Bray–Curtis distance and the binary Jaccard distance for β diversity. This showed that α diversity of the urinary bacterial community of NMIBC group was lower than those of the healthy control groups (Figure 1B,C). β diversity of the urinary microbiota evaluated by ANOSIM was significantly different between the two groups based on the Bray-Curtis distances (R = 0.144, *p* = 0.025, Figure 1D) and the binary Jaccard distance (R^2^ = 0.190, *p* = 0.003, Figure 1E). Then, the difference in bacterial community composition was analyzed. At the phylum, class, and genus level, the predominantly abundant phyla of the NMIBC group were *Eubacterium* sp. *CAG:581*, *Bacteroides* sp. *4_3_47FAA*, and *Flavobacteriales*, while the enriched phyla of the healthy control group were listed in the cyan columns in Figure 1F.

The functional composition of the urinary microbiome was compared between NMIBC patients and controls by COG and KEGG pathway analyses. Although the functional compositions of the two groups were highly similar, COG analyses indicated that the clustering of metabolic modules was increased in the NMIBC group, including the metabolism of xenobiotics by cytochrome P450 (*E-*value = 31.357, *p* < 0.001), purine (*E-*value = 30.835, *p* = 0.022), flavone, and flavanol biosynthesis (*E-*value = 29.663, *p* = 0.029) (Figure 1G). KEGG pathway analysis showed that these differential metabolisms were mainly concentrated on oxidative phosphorylation, cardiac muscle contraction, and carbon metabolism (Figure 1H). These findings are consistent with the hypermetabolic activity and aggressive characteristics of NMIBC.

### 3.2. Coculture of Eubacterium *sp.* CAG:581 Promoted the Growth of NMIBC Organoids

Hypoxia is essential for the growth of obligate anaerobes, but prohibitive for the maintenance of viable tumor organoids [24]. To surmount this tradeoff in oxygen demands, we developed an NMIBC organoids and anaerobes coculture system, consisting of a normoxic apical chamber and a hypoxic basal chamber (Figure 2A). This showed that the size of the organoids on the chip increased over time, achieving a diameter of 30 μm within 21 days. The *Eubacterium* sp. *CAG:581* and *Bacteroides* sp. *4_3_47FAA* coculture groups, respectively, proliferated faster by 48.5% and 37.8% when compared with control NMIBC organoids on Day 21 (Figure 2B). On Day 21, bright-field images of *Eubacterium* sp. *CAG:581* coculture group presented heterogeneous morphologies, with a predominantly thin-walled cystic structure and a solid dense structure (Figure 2C). To gain insights into the underlying mechanisms, we randomly performed RNA-Seq to compare *Eubacterium* sp. *CAG:581*-cocultured NMIBC organoids (N14-N26) and control NMIBC organoids (N1-N13). It demonstrated that ECM1 and MMP9 were most significantly upregulated in *Eubacterium* sp. *CAG:581*-cocultured NMIBC organoids, while PTPN6 and IKZF3 were mostly downregulated in the heatmap and the ssGSEA analysis (Figure 2D,E), which should be further validated by molecular biological experiments.

### 3.3. Eubacterium *sp.* CAG:581 Activated ECM1/ERK1/2 Phosphorylation/MMP9 of NMIBC Organoids

Based on the above RNA-Seq predictive data, we have determined the transcriptional levels of *ptpn6*, *ikzf3*, *ecm1*, and *mmp9* in NMIBC organoids cocultured with *Eubacterium* sp. *CAG:581* and control NMIBC organoids using RT-PCR. It showed that the increased colony forming units (CFU) gradient (5 × 10^5^, 10^6^, 5 × 10^6^, 10^7^, 5 × 10^7^) of *Eubacterium* sp. *CAG:581* have significantly increased mRNA levels of *ecm1* and *mmp9* (Figure 3A). Then, we compared their protein expressions by the Western blotting assay, which also demonstrated an increased gradient of *Eubacterium* sp. *CAG:581* could upregulate the ECM1 and MMP9 of the NMIBC organoids as compared with the control organoids. It was reported that ECM1 induces tumor growth by promoting angiogenesis or enhancing the EGF signaling in the breast cancer [25]. We thus detected their expression of ERK1/2 phosphorylation and AKT phosphorylation, which manifested the upregulated ERK1/2 phosphorylation under the exposure of the increased CFU gradient of *Eubacterium* sp. *CAG:581* (Figure 3B). Then we used the shRNA of ECM1 to model t BCa organoid^shECM1^ and the control BCa organoid^vector^. When exposed to 10^7^ cfu *Eubacterium* sp. *CAG:581*, BCa organoid^shECM1^ was determined to have decreased expression of ERK1/2 phosphorylation and MMP9. Meanwhile, BCa organoid^shECM1^ and BCa organoid^vector^ were treated with 10 μM ravoxertinib (Rav) or ulixertinib (Uli), both of which are inhibitors of ERK1/2 [26,27]. The results showed that both Rav and Uli could impair the expression of MMP9 (Figure 3C). Lastly, we detected the proliferative size of Rav-treated or Uli-treated BCa organoid^shECM1^ and control BCa organoid^vector^ under the exposure to 10^7^ cfu *Eubacterium* sp. *CAG:581* for 21 days. We found that ERK1/2 inhibition or shECM1 could effectively prohibit the growth of BCa organoids (Figure 3D), which suggested that *Eubacterium* sp. *CAG:581* progressed NMIBC organoids by activating the ECM1/ERK1/2 phosphorylation/MMP9.

### 3.4. Eubacterium *sp.* CAG:581 Was Endowed with the Diagnostic Predictor for NMIBC

Increasing evidence points to the urinary microbiota as a possible key susceptibility factor for early-stage bladder cancer (BCa) progression [11,28]. However, its conclusive interpretation is often insufficient, given that various environmental conditions could significantly alter the regulation of urinary microbiota. Herein, we have evaluated the relationship between the amount of *Eubacterium* sp. *CAG:581*, ECM1, MMP9, and other baseline features in 51 NMIBC patients and 47 healthy controls (Figure 4A). The amount of *Eubacterium* sp. *CAG:581* was positively associated with the occurrence of NMIBC (HR: 4.21, 95% CI: 2.54–5.33, log-rank *p* < 0.001). The expression of ECM1 (HR: 1.87, 95% CI: 1.61–2.83, log-rank *p* = 0.005) and MMP9 (HR: 1.66, 95% CI: 1.49–1.88, log-rank *p* = 0.013) were substantially higher in the NMIBC group than in the healthy control group. In contrast, age, sex, smoking status, and alcohol consumption were not statistically significant in this study for Pearson correlation analysis (Figure 4A). In addition, the linking of *Eubacterium* sp. *CAG:581*, ECM1, and MMP9 with survival probability of NMIBC were also analyzed by the Kaplan–Meier Curve. The findings revealed that higher amounts of *Eubacterium* sp. *CAG:581* were strongly associated with decreased survival probability in NMIBC (Figure 4B), and the survival time was also favorably linked with the expression of ECM1 and MMP9 (Figure 4C,D). The above data implied that the detection of *Eubacterium* sp. *CAG:581* is effective at predicting the occurrence of NMIBC.

### 3.5. Identification of NMIBC Occurrence-Associated Eubacterium *sp.* CAG:581 in the Larger Population

To further validate whether *Eubacterium* sp. *CAG:581* levels had a similar prediction value in a different and larger NMIBC patient population, we analyzed an additional cohort with 406 NMIBC patients as Cohort 2. Using PCA cluster analysis with 398 normal healthy controls, we identified the ability of *Eubacterium* sp. *CAG:581* to distinguish NMIBC urine from healthy control urine (Figure 5A). LASSO regression analysis of *Eubacterium* sp. *CAG:581*-induced prognostic DEGs was also modeled to verify its fine cooperativity (Figure 5B) and stable partial likelihood deviation from the minimum value (Figure 5C). Based on the abundance of *Eubacterium* sp. *CAG:581* levels, we have also divided them into high and low risk groups, which demonstrated that the prognosis model is feasible in their survival predictions (Figure 5D,E). The prognostic value of increased urine ECM1 and MMP9 was stable and accurate in a larger NMIBC cohort. Decreased PTPN6 and IKZF3 were determined in Cohort 2 as well (Figure 5F). The total survival rate was significantly worse (*p* < 0.001) in NMIBC patients with high amounts of *Eubacterium* sp. *CAG:581*, according to the Kaplan–Meier survival curve (Figure 5G). Receiver operating characteristic (ROC) curve analysis was conducted to predict the potential bladder cancer diagnosis using 1-, 2-, and 3-year NMIBC occurrence, and 3-year occurrence showed the highest AUC value of 0.79 (*p* value = 0.006). Therefore, this suggests a 79% chance that high *Eubacterium* sp. *CAG:581* will correctly distinguish an NMIBC patient from a healthy person within 3 years. The Youden Index was used to determine the optimal cut-off point of 3-year NMIBC occurrence as 10.3 (delta CT value), which provided the best balance between the sensitivity and the specificity of *Eubacterium* sp. *CAG:581* to predict NMIBC occurrence (Figure 5H). Therefore, the data in Cohort 2 not only confirmed our observations in Cohort 1, but also defined the potential value of the *Eubacterium* sp. *CAG:581* signature in predicting NMIBC occurrence.

## 4. Discussion

Several studies have demonstrated that 70% of NMIBC patients exhibit the disease progression for recurrence within 5 years, and 10–20% can progress to advanced muscle-invasive disease or distant metastatic disease [29]. In the case of an advanced stage, high-risk NMIBC patients are thus treated with electrodesiccation of the urinary bladder tumor and postoperative instillation of Bacillus Calmette–Guérin (BCG) [30]. Unfortunately, the instillation of BCG failed in approximately half of these patients, resulting in an approximately 30% persistence or early relapse for high-risk NMIBC. A total of 50% of them will undergo radical cystectomy [31]. Despite the advent of minimally invasive surgery and robotic technology, 90-day mortality and morbidity rates for patients undergoing cystectomy still remained as high as 3–6% and 28–64%, respectively [32]. For the past 30 years, bladder perfusion for NMIBC patients has been dominated by chemotherapeutic agents in China, and BCG immunotherapy in the United States. However, bladder chemotherapy instillation is an invasive drug delivery method which has led to a 30% occurrence of urethral irritation such as dysuria, frequency, urgency, hematuria, and cystitis. Moreover, the treatment course usually took more than one year, resulting in poor patient compliance [33]. The standard treatment regimen of BCG, on the other hand, can trigger more adverse reactions. Not only is the prognosis relatively poor, but patients will also endure much suffering in the process. Currently, NMIBC diagnosis mainly relies on cystoscopy [34,35]. Conventional imaging methods (CT urography, IVU, or ultrasound, etc.) depend decisively on the personal experience of the physician, while ultrasonography, intravenous pyelogram, cystography, magnetic resonance imaging, and lymphography are not very sensitive. Cystoscopy is relatively sensitive, but invasive and expensive, which makes large-scale screening difficult [36]. Urine exfoliative cytology is highly specific, but lacks sensitivity [37,38]. Eurovision fluorescence in situ hybridization has high sensitivity, and is therefore widely used for the routine clinical detection of NMIBC, but exhibits poor sensitivity for low-grade or small tumors [39]. Although some studies suggest that methylated CpG sites in urine may be promising markers for the detection or monitoring of NMIBC [40], these assays are not fully adopted in routine clinical practice and still need to be validated in multicenter and large-scale cohorts in Asia. Novel insights for early-stage screening of NMIBC are still required.

In this study, we have compared 51 patients with NMIBC and 47 healthy controls as Cohort 1 to explore the clinical relevance of altered urinary microbiota with the development of NMIBC. Then, the functional composition of the urinary microbiome was compared between NMIBC patients and controls by COG and KEGG pathway analysis. *Eubacterium* sp. *CAG:581* was confirmed to promote the growth of NMIBC organoids by activating ECM1/ERK1/2 phosphorylation/MMP9. Lastly, we used multivariate Cox modeling to investigate the clinical relevance of urinary *Eubacterium* sp. *CAG:581* abundance with the prognosis of 406 NMIBC patients as Cohort 2, which suggested that urinary *Eubacterium* sp. *CAG:581* could be used for the early diagnosis and prognosis of NMIBC. Therefore, the phosphorylation of ERK1/2 signaling could regulate numerous transcription factors, which are known to activate the transcription of a broad number of MMPs. Herein, we found that inhibiting ERK1/2 phosphorylation resulted in decreased MMP-9 expression, indicating the mediation of ERK1/2 in promoting NMIBC progression. A similar study performed by Lee et al. showed tandem decreases in MMP-9 mRNA levels following the inhibition of ERK1/2 phosphorylation in rhabdomyosarcoma, fibrosarcoma, bladder, colon, and prostate carcinoma cells [41], which is in accordance with the results for the NMIBC patients of this study. In addition, as the amount of *Eubacterium* sp. *CAG:581* has influenced the expression of ERK1/2 phosphorylation and MMP9, ECM1 is also positioned as a potential therapeutic target for the functional transcriptional network with *Eubacterium* sp. *CAG:581*-induced NMIBC migration and invasion.

The normal flora of microbial colonies has developed over a long historical evolutionary process and reside both outside and inside the human body. Microbial flora exists in the oral cavity, urinary tract, and intestinal tract. These florae have been found to have a tremendous effect on their hosts. Intestinal flora has been widely used in the early-stage diagnosis of neurodegenerative diseases, mood disorders, and lung diseases, as well as in predicting the recurrence of digestive tract cancers [42,43,44]. In recent studies, a wealth of research on urinary flora has confirmed that bacteria in urine may play an essential role in many urinary disorders, such as urge incontinence [45]. The study of Thomas White et al. found that bacterial diversity of urinary flora was associated with high body mass index, hormonal status, and symptoms of urge incontinence [46]. Several studies of bioinformatics analysis have also shown that among NMIBC patients, the relative abundance of *Sphingomonas*, *Immunobacterium*, *Aeromonas*, and *Bacillus* spp. in the urinary flora of the tumor group was significantly higher than in the control group [11,47]. However, more detailed underlying validation is required for these predictions. In this study, MMP9 served as the downstream executor of NMIBC growth. It belongs to the family of zinc-dependent endopeptidases with proteolytic activity against extracellular matrix components and involved in numerous physiological and pathological processes, including tissue remodeling, embryonic development, tumor growth, and metastasis [48,49]. Previous studies have revealed that MMP-9 is involved into the whole process of pathogenesis in bladder cancer [48,50,51]; however, the present study has only focused on the occurrence of NIMBC. We are constructing a cohort to compare NMIBC patients and MIBC patients for further validating the *Eubacterium* sp. *CAG:581* signature in the conversion from NMIBC into MIBC.

## 5. Conclusions

In conclusion, understanding the mechanisms contributed by urine bacterium to NMIBC development has the potential to improve its early-stage diagnostic decisions. This study presents a novel mechanism of *Eubacterium* sp. *CAG:581* in upregulating the ECM1–MMP9 axis and identifies *Eubacterium* sp. *CAG:581* to be clinically useful in predicting the occurrence of NMIBC.

## Figures and Tables

**Figure 1 cancers-15-00809-f001:**
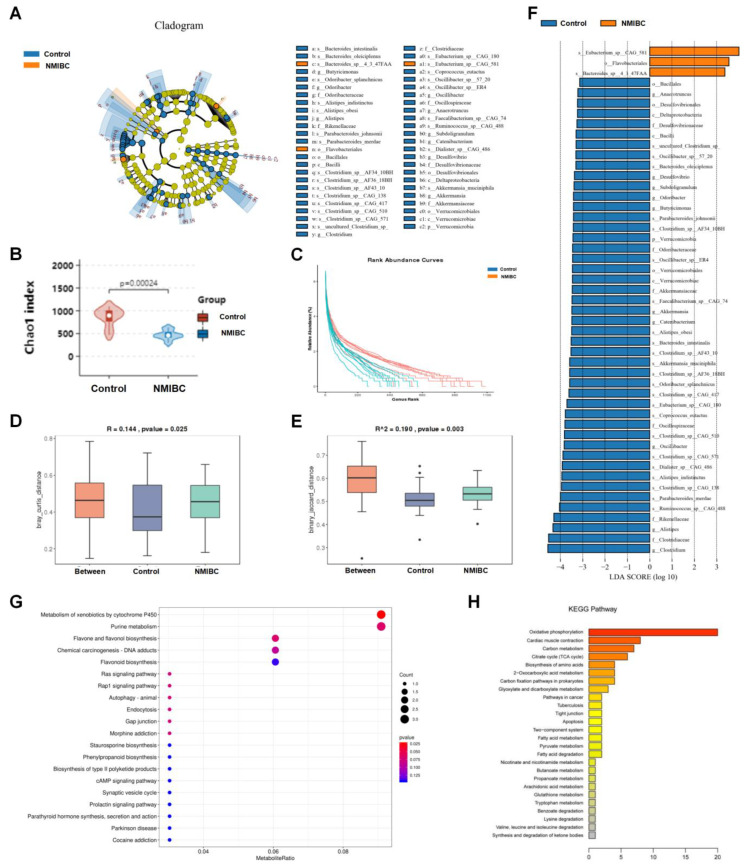
Metagenomics data manifested the clinical association between *Eubacterium* sp. *CAG:581* and NMIBC occurrence. (**A**) A cladogram representation of data in 51 NMIBC patients versus 47 healthy controls by 16S rDNA sequencing; taxa enriched in NMIBC patients (amber) and healthy controls (cobalt). The brightness of each dot is proportional to its effect size. (**B**,**C**) Chao 1 richness index (**B**) and rank abundance curves (C) of urine samples from 51 NMIBC patients (amber) versus 47 healthy controls (cobalt). (**D**,**E**) Bray–Curtis’s distance (**D**) and binary Jaccard distance (**E**) of urine samples from 51 NMIBC patients (amber) versus 47 healthy controls (cobalt). (**F**) Linear discriminant analysis (LDA) coupled with the effect size measurements identify the significant abundance of data in A. Taxa enriched in NMIBC patients are indicated with negative (cobalt) or positive (amber) LDA scores as compared with healthy controls, respectively. Only taxa greater than an LDA threshold of 3.5 are shown. (**G**) DEGs enriched in the COG classification. Dots represent term enrichment, with different colors based on Q values. Red color indicates high enrichment, while blue color indicates low enrichment. The sizes of the dots represent the enrichment of each gene. (**H**) Histogram presentation of the KEGG pathway. A total of 148 DEGs were successfully annotated and grouped into 26 functional categories. *p* values were determined using two-sided Fisher’s exact tests with Benjamini–Hochberg correction for multiple testing.

**Figure 2 cancers-15-00809-f002:**
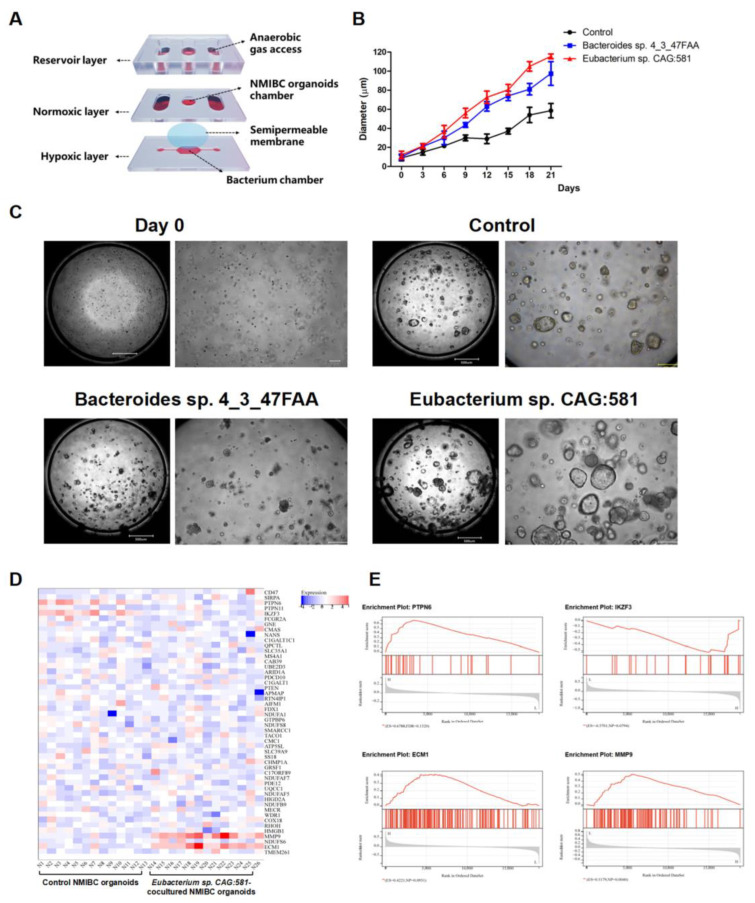
The coculture of *Eubacterium* sp. *CAG:581* promoted the growth of NMIBC organoids. (**A**) Schematic diagram of the coculture system of NMIBC organoids and anaerobic urine microbiota using the two-chambers chip. Each chip had a reservoir layer on the top, a 3D implanting hole in the middle, and an anaerobic culture slide underneath. (**B**) Growth curve of the NMIBC organoids cocultured with *Eubacterium* sp. *CAG:581* (10^6^ cfu per chip) and *Bacteroides* sp. *4_3_47FAA* (10^6^ cfu per chip). The results are presented as mean ± SD, *n* = 3. (**C**) Bright-field images showing the representative NMIBC organoid sizes on Day 0 and the *Eubacterium* sp. *CAG:581* coculture group, the *Bacteroides* sp. *4_3_47FAA* coculture group, and the NMIBC control group on Day 21. Scale bars, 1000 μm (left panels) and 10 μm (right panels). (**D**) Heatmap of DEGs of the *Eubacterium* sp. *CAG:581* coculture group as compared with the NMIBC control group on Day 21. Gene copy numbers are transformed as log2 ratios per gene (blue, decrease; red, increase). Gene expression profiling analysis was used to identify the potentially upregulated ECM1 and MMP9, as well as the downregulated PTPN6 and IKZF3 (|log2(Fold change)| > 1 and *p*-value < 0.05). (**E**) The Gene set enrichment analysis (GSEA) plot in the *Eubacterium* sp. *CAG:581* coculture group versus the NMIBC control group. The GSEA algorithm was used to evaluate the statistical significance of ECM1, MMP9, PTPN6, and IKZF3, according to the normalized enrichment scores (NES). These values were assigned to each gene and set after normalization across all analyzed gene sets.

**Figure 3 cancers-15-00809-f003:**
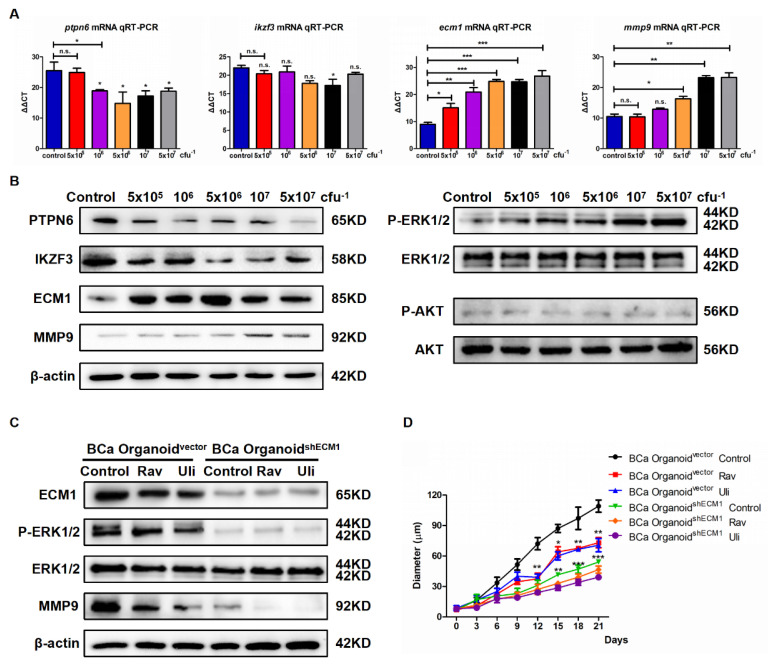
*Eubacterium* sp. *CAG:581* activated the ECM1/ERK1/2 phosphorylation/MMP9 of NMIBC organoids. (**A**) qRT-PCR analysis of *ptpn6, ikzf3, ecm1*, and *mmp9* levels of NMIBC organoids cocultured with increased an gradient (5 × 10^5^, 10^6^, 5 × 10^6^, 10^7^, 5 × 10^7^ cfu) of *Eubacterium* sp. *CAG:581*. (**B**) Western blot analysis of PTPN6, IKZF3, ECM1, MMP9, ERK1/2 phosphorylation, and AKT phosphorylation in NMIBC organoids cocultured with an increased gradient (5 × 10^5^, 10^6^, 5 × 10^6^, 10^7^, 5 × 10^7^ cfu) of *Eubacterium* sp. *CAG:581*. (**C**) Western blot analysis of ECM1, MMP9, and ERK1/2 phosphorylation in BCa organoid^shECM1^ and BCa organoid^vector^ treated with 10 μM ravoxertinib (Rav) or ulixertinib (Uli). (**D**) Growth curve of BCa organoid^shECM1^ and BCa organoid^vector^ treated with 10 μM ravoxertinib (Rav) or ulixertinib (Uli). Results are presented as mean ± SD, *n* = 3. * *p* < 0.05, ** *p* < 0.01, *** *p* < 0.001.

**Figure 4 cancers-15-00809-f004:**
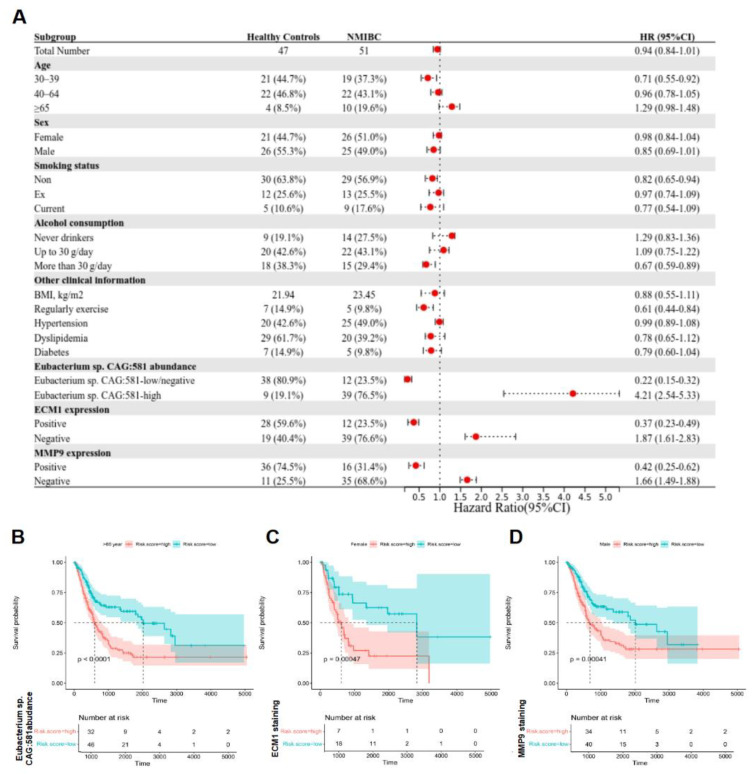
*Eubacterium* sp. *CAG:581* was endowed with the diagnostic predictor for NMIBC. (**A**) Univariate analysis of baseline information was performed in 51 NMIBC patients versus 47 healthy controls. The bars correspond to 95% confidence intervals. (**B**) Univariate predictive value of *Eubacterium* sp. *CAG:581* was compared between 51 NMIBC patients and 47 healthy controls, log-rank test. (**C**) Univariate predictive value of ECM1 was compared between 51 NMIBC patients and 47 healthy controls, log-rank test. (**D**) Univariate predictive value of MMP9 was compared between 51 NMIBC patients and 47 healthy controls, log-rank test.

**Figure 5 cancers-15-00809-f005:**
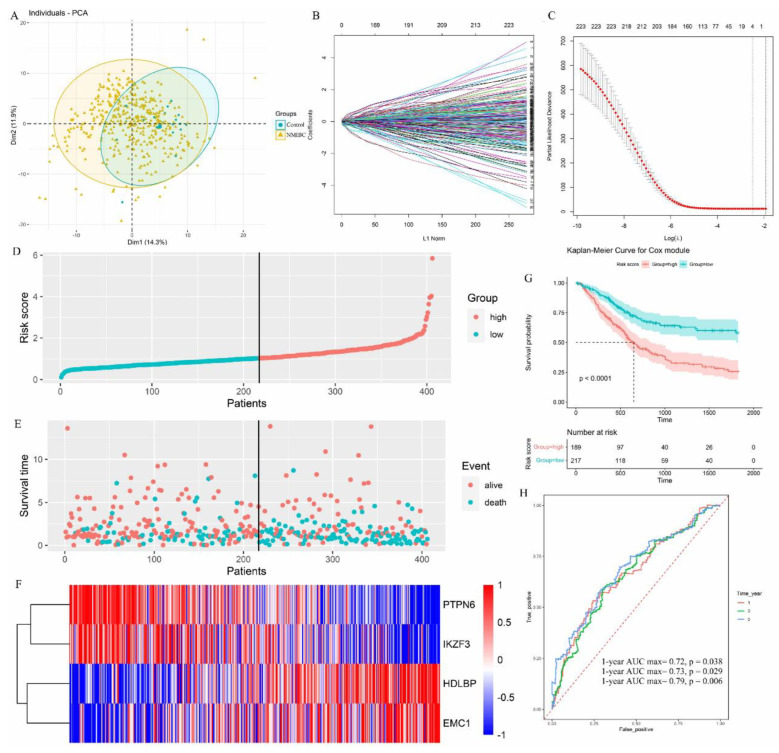
Identification of NMIBC occurrence-associated *Eubacterium* sp. *CAG:581* signature in the larger population. (**A**) The result of PCA analysis demonstrating the efficacy of the *Eubacterium* sp. *CAG:581* signature to distinguish NMIBC from the healthy control. (**B**) LASSO coefficient profiles of the prognostic DEGs. Each curve represents a coefficient, and the *x*-axis represents the regularization penalty parameter. As the tuning parameter (λ) changes, a coefficient that becomes non-zero enters the LASSO regression model. (**C**) Cross-validation to select the optimal λ. The red dotted vertical line crosses over the optimal log λ, which corresponds to the minimum value for multivariate Cox modeling. The two dotted lines represent one standard deviation from the minimum value. (**D**) The distribution of risk scores is shown for Cohort 2. The dotted horizonal line indicates the cut-off level of the risk score used to stratify patients, and the dotted vertical line separates participants on the basis of low risk (green) or high risk (red). (**E**) The distribution of overall outcomes in Cohort 2. NMIBC patients with a low-abundance of *Eubacterium* sp. *CAG:581* are shown in green, while NMIBC patients with high-abundance of *Eubacterium* sp. *CAG:581* are shown in red. (**F**) Heatmap of ECM1, MMP9, PTPN6, and IKZF3 in Cohort 2. Gene copy numbers are transformed as log2 ratios per gene (blue, decrease; red, increase). (**G**) Kaplan–Meier survival plots to predict patients at risk for NMIBC occurrence in Cohort 2. The number of patients remaining at a particular timepoint is shown at the bottom. (**H**) Time-dependent ROC curves for predicting one-year, two-year, and three-year survival in Cohort 2, with the AUC values of 0.72, 0,73, and 0.79, respectively. ROC, receiver operating curve; AUC, area under the curve.

## Data Availability

All generated, as well as analyzed, data in the present research are contained in this manuscript.

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
