# Peer review of "Urinary Eubacterium sp. CAG:581 Promotes Non-Muscle Invasive Bladder Cancer (NMIBC) Development through the ECM1/MMP9 Pathway"

_cancers, 2023, doi:10.3390/cancers15030809_

Round 1

Reviewer 1 Report

Comments on Cancers-2092900

In this manuscript, Yuhang Zhang et al. reported a potential association between urinary Eubacterium sp. CAG:581 and NMIBC progression by metagenomics analysis of clinical samples. With the comparison of NMIBC and healthy controls, the authors suggested that Eubacterium sp. CAG:581 might serve as a potential diagnostic predictor of NMIBC, and further investigated its cut-off value for clinical application. Overall, the findings are informative for future clinical validations, while a few points should be addressed properly before being considered for publication.

1.      The basic information (origin, passage, detailed culture conditions, etc) of the primary cells used for organoid culture was missing.

2.      The information of patient samples used in RNA-seq was missing, were they isolated freshly, frozen or extracted from FFPE blocks?

3.      Were the NMIBC patients undergoing any antibiotic treatments when the samples got collected?

4.      Is there any correlation between Eubacterium sp. CAG:581, as well as identified ECM1/MMP9/PTPN6/IKZF3 and the disease severity/aggressiveness/invasiveness in NMIBC patients? A representative IHC/IF staining should be included.

5.      Was the Eubacterium sp. CAG:581-induced disease progression in organoids shown to be alleviated by antibiotic treatment? A treatment group could help to strengthen the statement.

6.      Fig 2D: please assign N1-N26 into healthy/NMIBC groups.

7.      The current evidence about “Eubacterium sp. CAG:581 progressed NMIBC organoids by activating ECM1/ERK1/2 phosphorylation/MMP9” may not be sufficient, a more comprehensive pathway analysis would be required.

8.      Cohort 1, the manuscript mentioned there were 51 NMIBC and 47 healthy controls included, while in Fig 4A and supplemental data the numbers appeared to be inverted.

Author Response

January 11th, 2023

Point-by-point response to cancers-2092900

Thank you for the careful review of our paper entitled “Urinary Eubacterium sp. CAG:581 promotes non-muscle-invasive bladder cancer (NMIBC) development through ECM1/MMP9 pathway”. We really appreciate your constructive and thoughtful comments on this manuscript. Your insightful comments are not only valuable and helpful for improving this manuscript, but also of guiding significance to our future studies. Following these useful suggestions, we have amended the methods details and revised this manuscript in its graphical presentation and textual descriptions. The point-by-point response of your constructive comments are enumerated as below:

Reviewer #1 comments:

In this manuscript, Yuhang Zhang et al. reported a potential association between urinary Eubacterium sp. CAG:581 and NMIBC progression by metagenomics analysis of clinical samples. With the comparison of NMIBC and healthy controls, the authors suggested that Eubacterium sp. CAG:581 might serve as a potential diagnostic predictor of NMIBC, and further investigated its cut-off value for clinical application. Overall, the findings are informative for future clinical validations, while a few points should be addressed properly before being considered for publication.

Answer: Thank you for your encouraging comments on the clinical applicability and pertinent suggestions to revise this manuscript.

  1. The basic information (origin, passage, detailed culture conditions, etc) of the primary cells used for organoid culture was missing.

Answer: Thank you for the suggestion. The primary organoids establishment details were added as follows: Surgically resected tissues were obtained from patients diagnosed with NMIBC in cohort 1, which were collected endoscopically by cold cup biopsy. Then all NMIBC tissue samples were placed on ice in the advanced Dulbecco’s Modified Eagle Medium/Nutrient Mixture (DMEM) F‐12 medium and digested for 45 min at 37°C. The remaining tissue aggregates were further digested in 4 mL of TrypLE Express recombinant enzyme (Invitrogen) for another 5 min at 37°C. Subsequently, the suspension was filtered through a 70‐μm nylon cell strainer and centrifuged for 5 min at 300 g. The tissue pellets were suspended in cold Matrigel, and Matrigel cell suspension were seeded in prewarmed 6‐well culture plates for passaging. Each organoid line was used from passage 5 to 12 (in most cases at passage 8 or 9), so that lines displaying phenotypic instability had already completed their shift to a basal phenotype.

  1. The information of patient samples used in RNA-seq was missing, were they isolated freshly, frozen or extracted from FFPE blocks?

Answer: Thank you for your question. We have revised the sample information for sequencing analysis as follows: On the one hand, urine samples and formalin-fixed paraffin-embedded tissues (FFPE) of 51 NMIBC patients and 47 healthy controls were collected as the Cohort 1. We performed metagenome sequencing studies for the urine samples of Cohort 1 to define which bacterium is predominant in the urine of NMIBC patients as compared to those of healthy controls. Then their FFPE were isolated with RNA for RNA-Sequencing. On the other hand, we used the urine samples of 406 NMIBC patients and 398 healthy controls in Cohort 2 as a validation dataset to determine which levels of urinary Eubacterium sp. CAG:581 generated from Cohort 1 could be applied and validated. Thus, Cohorts 1 and 2 serve as the models for our research discovery and validation, respectively.

  1. Were the NMIBC patients undergoing any antibiotic treatments when the samples got collected?

Answer: Thank you for the question. To standardize the results of metagenome sequencing, herein we have selected NMIBC patients who had not received any antibiotic treatment prior to samples collection, for eliminating the possible effect of different levels of antibiotic treatments. The process of NMIBC primary organoids culture was also eradicated with antibiotic exposure.

  1. Is there any correlation between Eubacterium sp. CAG:581, as well as identified ECM1/MMP9/PTPN6/IKZF3 and the disease severity/aggressiveness/invasiveness in NMIBC patients? A representative IHC/IF staining should be included.

Answer: Thank you for this insightful question. As we have analyzed the limitation in our discussion section that “Previous studies have revealed that MMP-9 got involved into the whole process of pathogenesis in bladder cancer [45, 47, 48]; however, the present study has only focused on the occurrence of NIMBC. We are constructing the cohort to compare NMIBC patients and MIBC patients for further validating the Eubacterium sp. CAG:581 signature in the conversion from NMIBC into MIBC”, this study mainly focused to compare the healthy cohort and NMIBC patient and identify the key urine bacteria as the biomarker for early screening of bladder cancer in large scale. But in our subsequent cohort study, we would specially investigate the potential urine microbiota and underlying mechanism of disease severity/aggressiveness/invasiveness from NMIBC to MIBC.

  1. Was the Eubacterium sp. CAG:581-induced disease progression in organoids shown to be alleviated by antibiotic treatment? A treatment group could help to strengthen the statement.

Answer: Thank you for the question. Since no specific antibiotics for Eubacterium sp. CAG:581 inhibition was reported up till now, we have not designed antibiotic treatment group to alleviate tumor progression in organoids for validation.

  1. Fig 2D: please assign N1-N26 into healthy/NMIBC groups.

Answer: Thank you for the suggestion. We have added the annotations of control NMIBC organoids (N1-N13) and Eubacterium sp. CAG:581-cocultured NMIBC organoids (N14-N26) into the Figure 2D.

  1. The current evidence about “Eubacterium sp. CAG:581 progressed NMIBC organoids by activating ECM1/ERK1/2 phosphorylation/MMP9” may not be sufficient, a more comprehensive pathway analysis would be required.

Answer: Thank you for the suggestion. We have to acknowledge that Eubacterium sp. CAG:581 may contribute to NMIBC occurrence by more than one pathway and ECM1/ERK1/2 phosphorylation/MMP9 is just one of them. Thus, the title of this part in results was revised as follows: Eubacterium sp. CAG:581 could activate ECM1/ERK1/2 phosphorylation/MMP9 to promote NMIBC organoids development.

  1. Cohort 1, the manuscript mentioned there were 51 NMIBC and 47 healthy controls included, while in Fig 4A and supplemental data the numbers appeared to be inverted.

Answer: Thank you for the suggestion. Sorry for uploading the wrong version of Figure 4 and we have corrected it.

In the submitted manuscript Zhang et al. showed that urinary Eubacterium sp. CAG:581 promoted non-muscle-invasive bladder cancer (NMIBC) progression through ECM1/MMP9 pathway, which may serve as the acceptable noninvasive diagnostic biomarker for NMIBC.

If additional work needs to be done, please feel free to contact us. Many thanks to all of your detailed and useful suggestions in revising this manuscript again.

Sincerely yours,

Professor Yi-Min Cui, MD, PhD

Department of Pharmacy, Peking University First Hospital, Beijing, China

[email protected]

Tel. 86-010-83950400

Reviewer 2 Report

In the submitted manuscript Zhang et al. showed that urinary Eubacterium sp. CAG:581 promoted non-muscle-invasive bladder cancer (NMIBC) progression through ECM1/MMP9 pathway, which may serve as the acceptable noninvasive diagnostic biomarker for NMIBC.

This manuscript is quite well written, hypothesis is scientifically sound, proper number of adequate experiments has been conducted, and conclusions were corroborated by results. However, the biggest drawback is insufficiently detailed description of methods because of which study behind this manuscript is to a great deal irreproducible!

1) All used methods must be described in "Materials and methods" section, not in "Results" or figure legends!

2) It is unclear which FFPE tissues were collected from NMIBC patients and healthy control, and for what actually those FFPE tissues were used?!

3) It is unclear how were urine samples collected, bacterial DNA isolated, or urinary microbiome separated from urine for bacterial RNA isolation?!

4) Precisely explain how were metagenome data analyzed, including all indexes, LEfSe, COG, KEGG, PCA, LASSO, etc. (Figure 1).

5) For EVERY bioinformatic tool used provide its version number and cite proper reference.

6) It is unclear which tissue was used for organoid creation, and how was it prepared. Provide O2 level during hypoxic condition.

7) Recheck text that all nonstandard abbreviations (e.g., MOI) were explained after first mentioning.

8) Seurat is used for scRNA-seq, not bulk RNA-seq data analysis. Therefore, it is unclear which RNA-seq method was actually used and which difference in expression was considered statistically significant!

9) Provide reference for 2^-ddCt method (DOI: 10.1006/meth.2001.1262).

10) Application of some statistical analysis is incomprehensible:

- Provide which post hoc test was used for ANOVA.

- Pearson correlation analysis was not mentioned in "Statistical Analysis".

- It is unclear why hazard ratio was used for comparing different variables between NMIBC and healthy control group (especially for Supplementary Table S1 and S2) when it was written that "Univariate and multivariate Cox regression models were used to evaluate survival. The hazard ratio (HR) and 95 percent confidence interval (CI) were adopted to find genes correlated with overall survival."?!

- Results must be presented with mean ± SD (using SEM could be form of cheating), and -SD error bar should be visible on ALL bar graphs (Figure 3A)!

- Provide p-values for AUC, and interpret AUC according to, e.g., DOI: 10.1097/JTO.0b013e3181ec173d

11) Rewrite "NMIBC organoids cocultured with urinary bacterium in the 2-chamber culture system" section in passive past tense - it should not be written like a recipe! Furthermore, treatment with ravoxertinib or ulixertinib were not explained in methods.

12) Western blot bands must be quantified (e.g., using ImageJ software) and properly statistically analyzed.

13) Provide qPCR cycling conditions.

14) Provide exact model and manufacturer of ALL used instruments (e.g., qPCR machine and WB imager).

15) Methodological parts of Results' section "Coculture of Eubacterium sp. CAG:581 promoted the growth of NMIBC organoids" put in "Materials and methods".

16) Supplementary Figure S1 was not cited not mentioned in the manuscript, while provided in supplementary.

Author Response

January 11th, 2023

Point-by-point response to cancers-2092900

Thank you for the careful review of our paper entitled “Urinary Eubacterium sp. CAG:581 promotes non-muscle-invasive bladder cancer (NMIBC) development through ECM1/MMP9 pathway”. We really appreciate your constructive and thoughtful comments on this manuscript. Your insightful comments are not only valuable and helpful for improving this manuscript, but also of guiding significance to our future studies. Following these useful suggestions, we have amended the methods details and revised this manuscript in its graphical presentation and textual descriptions. The point-by-point response of your constructive comments are enumerated as below:

Reviewer #2 comments:

This manuscript is quite well written, hypothesis is scientifically sound, proper number of adequate experiments has been conducted, and conclusions were corroborated by results. However, the biggest drawback is insufficiently detailed description of methods because of which study behind this manuscript is to a great deal irreproducible!

Answer: Thank you for your encouraging comments on the novelty and workload of this study. We have revised this manuscript especially for its methods section according to your pertinent suggestions as follows:

  1. All used methods must be described in "Materials and methods" section, not in "Results" or figure legends.

Answer: Thank you for the suggestion. We have moved the following several methodological elements from results into the section of methods as marked:

  • The plug of coculture system inserts tightly, physically blocking the influx of external oxygen, which allows maintenance of hypoxia in the basal chamber while oxygen freely perfuses the apical chamber. NMIBC organoids were established on the array chip using 50% Matrigel. Each chip had a reservoir layer on the top, a 3D implanting hole in the middle, and anaerobes culture slide underneath. The nested design allowed convenient medium exchange without disruption of the 3D organoids.
  • LEfSe scores measure the consistency of differences in relative abundance between taxa in the groups analyzed, with a higher core indicating higher consistency. We considered taxa with linear discriminant analysis score >2 and p < 0.05 to be significant. To identify species represented by the genera revealed by LEfSe, we first identified the OTUs associated with those genera, filtered low abundance OTUs (<50 copies), performed Kruskal-Wallis tests on each remaining OTU, and used BLAST (https://blast.ncbi.nlm.nih.gov) to align the sequences of these OTUs against the Greengenes database, retaining species with identity match of >97%. The BLAST algorithm was also used to query the predicted genes against the Kyoto Encyclopedia of Genes and Genomes (KEGG) database (http://www.genome.jp/kegg/), and the corresponding biological pathways were determined based on the obtained KEGG Orthology (KO) numbers. Gene Ontology (GO, http://www.geneontology.org/) annotations of the BLAST results were analyzed using blast2go. The genomic circle map was constructed using Circos v0.64 (http://circos.ca/).
  • When passaging to the 4-8th generation, BCa organoidshECM1 and control BCa organoidvector were exposed to 10 μM Ravoxertinib (Rav) or Ulixertinib (Uli) coculture for 24h to identify downstream pathway activation and 21 days for 107 cfu Eubacterium sp. CAG:581 of proliferation analysis. SCH772984, ravoxertinib and LY3214996, ulixertinib were both purchased from Selleckchem.
  1. It is unclear which FFPE tissues were collected from NMIBC patients and healthy control, and for what actually those FFPE tissues were used?

Answer: Thank you for the question. We have revised the information of sample use as follows: On the one hand, urine samples and formalin-fixed paraffin-embedded tissues (FFPE) of 51 NMIBC patients and 47 healthy controls were collected as the Cohort 1. We performed metagenome sequencing studies for the urine samples of Cohort 1 to define which bacterium is predominant in the urine of NMIBC patients as compared to those of healthy controls. Then their FFPE were isolated with RNA for RNA-Sequencing. On the other hand, we used the urine samples of 406 NMIBC patients and 398 healthy controls in Cohort 2 as a validation dataset to determine which levels of urinary Eubacterium sp. CAG:581 generated from Cohort 1 could be applied and validated. Thus, Cohorts 1 and 2 serve as the models for our research discovery and validation, respectively.

  1. It is unclear how were urine samples collected, bacterial DNA isolated, or urinary microbiome separated from urine for bacterial RNA isolation?

Answer: Thank you for the question. We have added it as follows: We collected urine samples primarily by collecting midstream clean capture urine (CC) and microbiologically cultured the urine using enhanced quantitative urine culture (EQUC) by inoculating 100 μL of urine onto 5% CO blood agar plates, mucin-nalidixic acid agar plates and MacConkey agar plates for 248 hours, followed by sequencing using 16S rRNA amplicons.

  1. Precisely explain how were metagenome data analyzed, including all indexes, LEfSe, COG, KEGG, PCA, LASSO, etc. (Figure 1).

Answer: Thank you for the suggestion. We have added their analysis process as follows: The LEfSe score measures the consistency of relative abundance differences among the taxa analyzed, and taxa with linear discriminant analysis scores > 2 and p < 0.05 were judged by python2 (lefse v20171228) as significant. We identified species represented by the genera revealed by LEfSe and filtered for low abundance (<50 copies), then performed the Kruskalvolis test for each remaining OTU, and used BLAST (https://blast.ncbi.nlm.nih.gov) to match the sequences of these OTUs to the Greengenes database, retaining species identity matches >97%. We queried the predicted genes in the Kyoto Encyclopedia of Genes and Genomes (KEGG) database (http://www.genome.jp/kegg/) using the BLAST algorithm and identified the corresponding biological pathways based on the obtained KEGG Orthology (KO) numbers. We have also analyzed Gene Ontology (GO, http://www.geneontology.org/) annotations of BLAST results using blast2go and constructed genomic circular maps using Circos v0.64 (http://circos.ca/), and used NCBI Blast to compare gene protein sequences with COG database for comparison and to obtain its functional annotation information.

  1. For every bioinformatic tool used provide its version number and cite proper reference.

Answer: Thank you for the suggestion. In this study, we have applied python2 (lefse v20171228) for LEfSe (Line Discriminant Analysis (LDA) Effect Size) analysis (PMID: 21702898), R v3.1.1 for Chao1, Ace, Shannon, Simpson analysis (picante, v1.8.2) (PMID: 34259548) and PERMANOVA/ANOSIM analysis (vegan,v2.3-0) (PMID: 34002024).

  1. It is unclear which tissue was used for organoid creation, and how was it prepared. Provide O2 level during hypoxic condition.

Answer: Thank you for the suggestion, and we have added details to the cocultured NMIBC organoids establishment as follows: Surgically resected tissues were obtained from patients diagnosed with NMIBC in cohort 1, which were collected endoscopically by cold cup biopsy. Then all NMIBC tissue samples were placed on ice in the advanced Dulbecco’s Modified Eagle Medium/Nutrient Mixture (DMEM) F‐12 medium and digested for 45 min at 37°C. The remaining tissue aggregates were further digested in 4 mL of TrypLE Express recombinant enzyme (Invitrogen) for another 5 min at 37°C. Subsequently, the suspension was filtered through a 70‐μm nylon cell strainer and centrifuged for 5 min at 300 g. The tissue pellets were suspended in cold Matrigel, and Matrigel cell suspension were seeded in prewarmed 6‐well culture plates for passaging. Each organoid line was used from passage 5 to 12 (in most cases at passage 8 or 9), so that lines displaying phenotypic instability had already completed their shift to a basal phenotype.

The plug of coculture system inserts tightly, physically blocking the influx of external oxygen, which allows maintenance of hypoxia in the basal chamber, while oxygen freely perfuses the apical chamber. NMIBC organoids were established on the array chip using 50% Matrigel. Each chip had a reservoir layer on the top, a 3D implanting hole in the middle, and anaerobes culture slide underneath. The nested design allowed convenient medium exchange without disruption of the 3D organoids. After the coating process for 3 days or when the layer is close to confluence, equilibrate the medium in the basal chamber with anaerobic gas and subsequently sealed by inserting a plug made of butyl rubber (AsONE international, Santa Clara, CA). The oxygen concentration of the apical chamber was measured by a fiberoptic oxygen meter (PreSens. Regensburg, Germany), which was set with 0mg/L-0.2mg/L oxygen. Then we added urinary bacterium to the basal side in a medium suitable for exposure. For multiplicity of infection (MOI) calculations, a representative well can be harvested and cells counted as described above.

  1. Recheck text that all nonstandard abbreviations (e.g., MOI) were explained after first mentioning.

Answer: Thank you for the suggestion, and we have added the abbreviation of MOI and checked to identify all abbreviations when they first appear.

  1. Seurat is used for scRNA-seq, not bulk RNA-seq data analysis. Therefore, it is unclear which RNA-seq method was actually used and which difference in expression was considered statistically significant.

Answer: Thank you for the question. Total RNA was isolated from FFPE of cohort 1 using the RNeasy Micro Kit, then they were converted to cDNA with poly A primers using a TruSeq RNA Sample Preparation kit v2 (Illumina). FPKM (fragments per kilobase of transcript per million mapped fragments) were calculated as gene expression level using Cufflinks version 2.2.1. High-throughput sequencing for mRNA-seq was carried out using a Hiseq2500 (Illumina) system. For analysis and visualization of the data generated by Cufflinks, we used the DEseq R package to perform differential analysis of gene expression, and the screening criteria were set as |log2(Fold change)|>1 and p-value< 0.05.

  1. Provide reference for 2^-ddCt method (DOI: 10.1006/meth.2001.1262).

Answer: Thank you for the suggestion, and we have provided this suggested reference into the new version.

  1. Application of some statistical analysis is incomprehensible:
  • Provide which post hoc test was used for ANOVA.

Answer: Thank you for the question, and we have used S-N-K (Student-Newman-Keuls) post hoc test in the analysis of ANOVA.

  • Pearson correlation analysis was not mentioned in "Statistical Analysis".

Answer: Thank you for the suggestion, and we have added it into the part of statistical analysis as follows: Mann-Whitney test and Pearson correlation and Chi-Square tests of independence are used for the correlation analyses.

  • It is unclear why hazard ratio was used for comparing different variables between NMIBC and healthy control group (especially for Supplementary Table S1 and S2) when it was written that "Univariate and multivariate Cox regression models were used to evaluate survival. The hazard ratio (HR) and 95 percent confidence interval (CI) were adopted to find genes correlated with overall survival."?

Answer: Thank you for the question, and we have revised this part as follows: Univariate and multivariate Cox regression models were used to evaluate the occurrence of NMIBC. The hazard ratio (HR) and 95 percent confidence interval (CI) were adopted to find key factors correlated with NMIBC occurrence.

  • Results must be presented with mean ± SD (using SEM could be form of cheating), and -SD error bar should be visible on all bar graphs (Figure 3A).

Answer: Thank you for the suggestion. We have replaced all figures of mean ± SEM with mean ± SD. The relative results descriptions and figure legends were also revised as marked.

  • Provide p-values for AUC, and interpret AUC according to, e.g., DOI: 10.1097/JTO.0b013e3181ec173d

Answer: Thank you for the suggestion. We have calculated the p-values for AUC curves of Figure 5H and added the descriptions as follows: Receiver operating characteristic (ROC) curve analysis was conducted to predict the potential bladder cancer diagnosis using 1-, 2-, and 3-year NMIBC occurrence and 3-year occurrence was demonstrated with the highest AUC value of 0.79 (p value = 0.006). It thus suggests an 79% chance that high Eubacterium sp. CAG:581 will correctly distinguish one NMIBC patient from healthy population within 3 years.

  1. Rewrite "NMIBC organoids cocultured with urinary bacterium in the 2-chamber culture system" section in passive past tense - it should not be written like a recipe! Furthermore, treatment with ravoxertinib or ulixertinib were not explained in methods.

Answer: Thank you for the suggestion, and we have revised it as follows: NMIBC organoids was cocultured with urinary bacterium in the 2-chamber culture system.

For the treatments of ravoxertinib and ulixertinib, we have added it as follows: When passaging to the 4-8th generation, BCa organoidshECM1 and control BCa organoidvector were exposed to 10 μM Ravoxertinib (Rav) or Ulixertinib (Uli) coculture for 24h to identify downstream pathway activation and 21 days for 107 cfu Eubacterium sp. CAG:581 of proliferation analysis. SCH772984, ravoxertinib and LY3214996, ulixertinib were both purchased from Selleckchem.

  1. Western blot bands must be quantified (e.g., using ImageJ software) and properly statistically analyzed.

Answer: Thank you for the suggestion. We have quantified all Western blot bands and statistically analyzed as Supplementary Figure S2.

  1. Provide qPCR cycling conditions.

Answer: Thank you for the suggestion. We have added more details about the reaction condition in this section of Quantitative Real-Time PCR that “Quantitative real‐time PCR was performed using the Hieff qPCR SYBR Green PCR Master Mix (Yeasen, China) on the ABI 7300 TH Real-Time PCR System (Applied Biosystems). 10 µL of dye, 1 µL of upstream and downstream primer, 1 µL of dNTP, 2 µL of Taq polymerase, 5 µL of cDNA of the sample to be tested and 30 µL of ddH2O were mixed and centrifuged briefly at 6000 rpm for PCR amplification reaction. The reaction conditions were: 2 min pre-denaturation at 93℃, then 40 cycles at 93℃ for 1 min, 55℃ for 1 min, 72℃ for 1 min, and finally 7 min extension at 72℃.”

  1. Provide exact model and manufacturer of all used instruments (e.g., qPCR machine and WB imager).

Answer: Thank you for the suggestion. We have checked and added all used instruments information as marked in the new version of methods.

  1. Methodological parts of Results' section "Coculture of Eubacterium sp. CAG:581 promoted the growth of NMIBC organoids" put in "Materials and methods".

Answer: Thank you for the suggestion. We have moved this part into the section of results as answered in the question 1.

  1. Supplementary Figure S1 was not cited not mentioned in the manuscript, while provided in supplementary.

Answer: Thank you for the suggestion. We have added the citations as follows: All primary images (Supplementary Figure S1) were cropped for presentation.

If additional work needs to be done, please feel free to contact us. Many thanks to all of your detailed and useful suggestions in revising this manuscript again.

Sincerely yours,

Professor Yi-Min Cui, MD, PhD

Department of Pharmacy, Peking University First Hospital, Beijing, China

[email protected]

Tel. 86-010-83950400

Round 2

Reviewer 1 Report

I'd thank the author for the illustration of most of the listed questions and recommend the paper to be accepted for publication.